# Pharyngeal mechanosensory neurons control food swallow in *Drosophila melanogaster*

**Jierui Qin[1,2†], Tingting Yang[1,2†], Kexin Li[1,2], Ting Liu[1], Wei Zhang[2]\***

[1]School of Life Sciences, IDG/McGovern Institute for Brain Research, Tsinghua University, Beijing, China; [2]Tsinghua-Peking Center for Life Science, Beijing, China

## eLife Assessment

This **valuable** study investigates the role of mechanosensory feedback during swallowing in adult *Drosophila*. The authors provide **convincing** evidence that three mechanotransduction channel genes are required for ingestion rhythms and localize the role of these genes to a specific subpopulation of pharyngeal mechanosensory neurons. However, there is incomplete evidence to support the conclusions that these sensory neurons are necessary for swallowing, respond to stretch during swallowing, and connect to the motor neurons that control swallowing. This work may be of interest to neuroscientists interested in motor control of feeding behavior.

**\*For correspondence:**
wei_zhang@mail.tsinghua.edu.cn

[†]These authors contributed equally to this work

**Competing interest:** The authors declare that no competing interests exist.

**Abstract** As the early step of food ingestion, the swallow is under rigorous sensorimotor control. Nevertheless, the mechanisms underlying swallow control at a molecular and circuitry level remain largely unknown. Here, we find that mutation of the mechanotransduction channel genes *nompC*, *Tmc*, or *piezo* impairs the regular pumping rhythm of the cibarium during feeding of the fruit fly *Drosophila melanogaster*. A group of multi-dendritic mechanosensory neurons, which co-express the three channels, wrap the cibarium and are crucial for coordinating the filling and emptying of the cibarium. Inhibition of them causes difficulty in food emptying in the cibarium, while their activation leads to difficulty in cibarium filling. Synaptic and functional connections are detected between the pharyngeal mechanosensory neurons and the motor circuit that controls swallow. This study elucidates the role of mechanosensation in swallow, and provides insights for a better understanding of the neural basis of food swallow.

## Introduction

Pharyngeal sensory organs serve as the final checkpoint before food ingestion, monitoring the chemical and physical properties of food (*Chen et al., 2019*; *Choi et al., 2016*; *Joseph et al., 2017*; *Kim et al., 2017*). Real-time sensory feedback generated during swallow peristalsis is vital to the process of expelling food bolus from the mouth to esophagus (*Manzo et al., 2012*; *McKellar, 2016*; *McKellar et al., 2020*; *Zhou et al., 2019*). The pharyngeal sensation can also impact the internal state and appetite of animals, while feeding status can in turn affect swallowing behavior (*Ryan, 2018*; *Saker et al., 2016*). In humans, tactile and pressure receptors on the tongue and palate receive sensory inputs from distributed areas (e.g. bolus texture, shape, and size) and relay information to the brain (*Iannilli et al., 2014*; *Simon et al., 2006*). Chewing and ingesting tough or viscous foods can cause increased 'oro-exposure time' and reduced food intake as a result (*Bolhuis et al., 2014*; *de Graaf, 2012*).

The pharynx contains a plethora of gustatory and mechanosensory neurons, and their functions are being gradually elucidated (*Chen and Dahanukar, 2017*; *Chen et al., 2019*; *Choi et al., 2016*; *Joseph et al., 2017*; *Kim et al., 2017*; *LeDue et al., 2015*). Physical properties of the food offer important information regarding its texture and influence animals' inclination to eat (*Torii and Moriyama, 2010*; *Wu et al., 2019b*). Although it has been extensively studied how the food texture is sensed by peripheral sensory organs in both *Drosophila* and other animals (*Li and Montell, 2021*; *Sánchez-Alcañiz et al., 2017*; *Zhang et al., 2016*), how the internal sensory neurons monitor the physical quality of food to regulate ingestion remains elusive.

Feeding behaviors such as chewing and sucking require rhythmic and coordinated contraction of muscle groups, and are thought to be controlled by central pattern generators (CPGs). CPGs have been proposed to control many complex motor behaviors in different systems. They are defined as a neural assembly by their intrinsic oscillation and independence from sensory input (*Grillner, 2006*; *Marder and Bucher, 2001*). In mammals, the muscles that coordinate food swallow are controlled by the CPGs in the brain stem (*Marder and Bucher, 2001*; *Amirali et al., 2001*). However, the identity of CPGs and their downstream circuits remain poorly understood.

In *Drosophila*, the swallow is driven by food pumping induced by the suction and compression of the cibarium (*Ferris, 1950*). Pump frequency is independent of the concentration of sucrose or feeding state, but is largely regulated by food viscosity (*Manzo et al., 2012*). Silencing of certain groups of motor neurons (MNs) could disrupt pumping and ingestion, while activation could elicit arrhythmic pumping (*Manzo et al., 2012*). Rhythmic fluid ingestion has also been studied in other insects (*Marder and Bucher, 2007*), but the molecular and neural basis controlling ingestion is still undiscovered.

Here, we identify a group of multi-dendritic mechanosensory neurons in the fly's cibarium (md-C neurons) which are essential for swallow control. Inhibition of these neurons leads to difficulty in cibarium emptying and lower ingestion efficiency, while activation of them causes higher pump frequency and sometimes difficulty in cibarium filling. md-C neurons interact with the MNs in the brain to control swallowing. Our work provides insights into the regulation mechanism of swallow.

## Results

### Mechanotransduction channel genes are required for swallowing behavior

Flies exhibit a rhythmic swallow pattern when ingesting food (*Ferris, 1950*; *Manzo et al., 2012*; *Figure 1A*). The whole action is composed of two steps: food is sucked into the cibarium (filling) and then expelled into the foregut (emptying). The frequency is not influenced by the taste but the viscosity of the food (*Manzo et al., 2012*), suggesting that the mechanical force exerted by food bolus passing the cibarium is essential in maintaining the swallow rhythm. We thus tested the swallowing behavior of the mutant alleles of three mechanotransduction channel genes (*nompC*, *piezo*, *Tmc*), and found that these flies exhibited a lower pump frequency compared to control groups. Disruption of the mechanoreceptor function had a profound impact on the emptying phase of swallowing (*Figure 1B–D*, *Video 1*), with filling and emptying taking longer and the filling/emptying time ratio decreasing significantly (*Figure 1—figure supplement 1A–C*). About one-third of *nompC* mutants exhibited difficulty in cibarium emptying (emptying time>0.3 s), indicating the essential role of mechanotransduction channel genes in swallowing behavior. However, the *nompC* mutant flies did not show a significant impairment in feeding efficiency, likely due to an increased volume of each pump (*Figure 1C*). These findings suggest that mechanosensation mainly contributes to the emptying phase of swallowing.

We investigated the involvement of mechanoreceptors in regulating swallow frequency in response to food viscosity. We fed flies with a water solution containing varying concentrations of methylcellulose (MC) to increase viscosity and found that flies with mutated *Tmc* or *piezo* genes consistently exhibited incomplete emptying when fed with water solutions containing 1% MC or higher concentrations, while low proportion of *nompC* mutants and wild-type flies sporadically exhibited incomplete filling when fed with water solution of the same viscosity (*Figure 1—figure supplement 1E–H*, *Video 2*). These findings suggest that all three mechanoreceptors are necessary for sensing the swallowing process and providing feedback to the downstream motor circuit to regulate pump strength, while

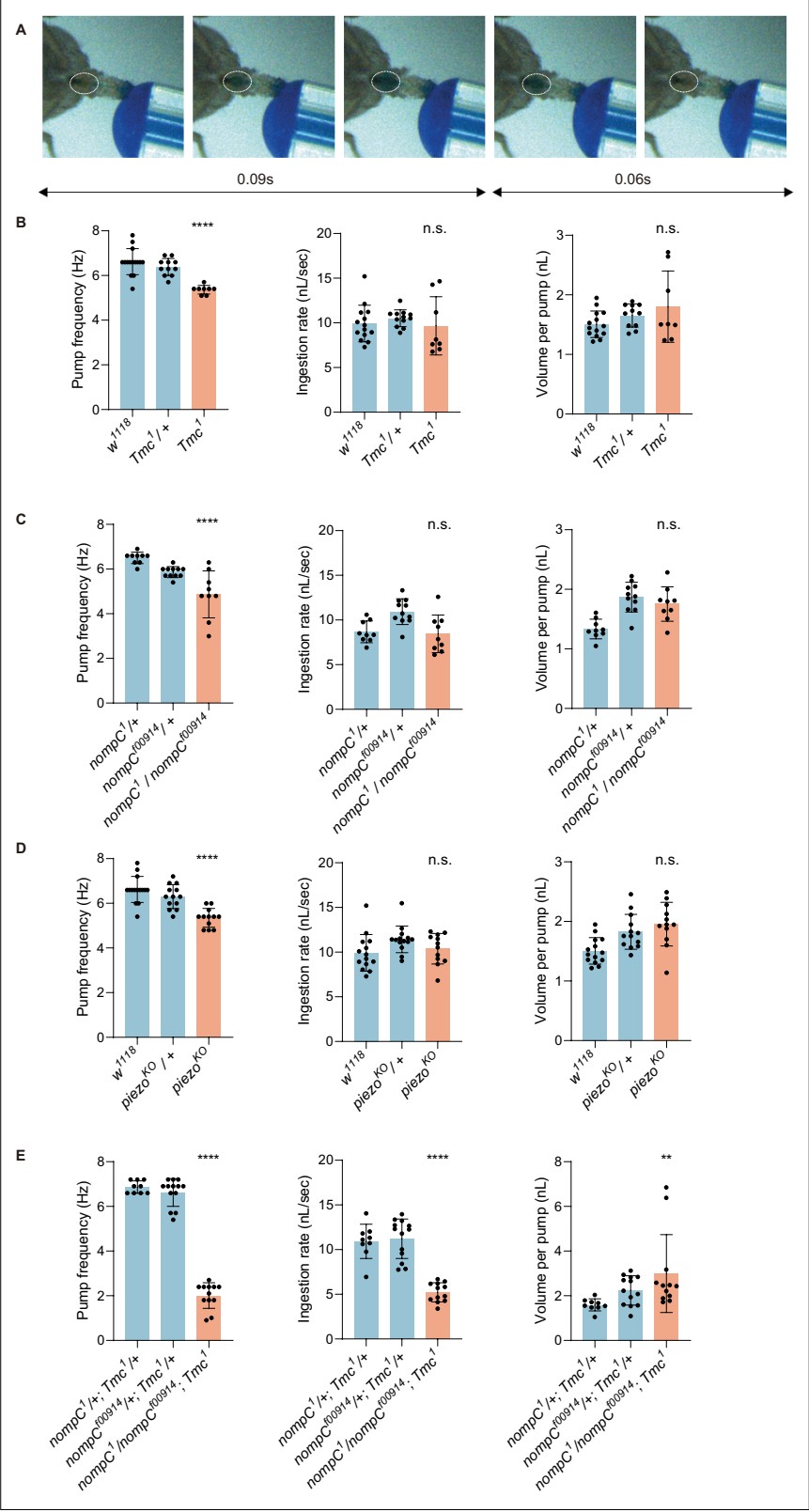

**Figure 1.** Mechanoreceptor genes are essential for swallow control of *Drosophila*. (**A**) Swallow patterns of liquid food. Filling and emptying process constitutes a cycle. (**B–D**) Swallowing behavior of *Tmc*, *nompC,* and *piezo* mutant flies. Pump frequency represents the swallowing speed, ingestion rate indicates the efficiency of food intake, while volume per pump shows how much a fly ingests with one pump. N=8–15 in each group. (**E**) Double

*Figure 1 continued on next page*

*Figure 1 continued*

mutans of *Tmc* and *nompC* show more severe dysphagia. n=9–13 in each group. In all analyses, one-way analysis of variance (ANOVA) followed by Dunnett's test for multiple comparisons was used, and statistical differences were represented as follows: *p<0.05, **p<0.01, ***p<0.001, and ****p<0.0001. Data were represented as means ± SEM.

The online version of this article includes the following figure supplement(s) for figure 1:

**Figure supplement 1.** Mutation of mechanoreceptors leads to a lower pump frequency caused by longer emptying time.

for foods with certain viscosity, *Tmc* or *piezo* mutants pump might be unable to support complete cibarium emptying. We propose that *nompC* plays a role in initiation, while *Tmc* and *piezo* control the driving strength of swallowing.

Moreover, double mutation of *Tmc* and *nompC* led to a more severe emptying difficulty, with pump frequency and ingestion rate both decreasing significantly (*Figure 1E*) and food tends to stay in the pharyngeal area for a longer time (*Video 1*). As a result, a higher pump volume was detected, although the overall intake efficiency was decreased due to the incomplete swallow process (*Figure 1E*). In contrast, the double mutant of *Tmc* and *piezo* showed a pump frequency of about 5 Hz, similar to flies of the mutant of either gene (*Figure 1—figure supplement 1D*).

Based on our findings that *Tmc*, *nompC*, and *piezo* are important for swallowing, we reasoned that neurons co-expressing these genes may play an essential role in swallow control. So, we explored the expression pattern of each gene, or the intersection between each two of the three genes in the brain and in the cibarium, and identified a set of multi-dendritic sensory neurons in the cibarium (*Figure 2—figure supplement 1*, *Figure 2F*). In the brain, subesophageal zone (SEZ) was commonly labeled by either driver, while *Tmc-GAL4* and *nompC-QF* intersection shows a clear projection pattern (*Figure 2F*). And by expressing nuclear-localized RFP, RedStinger, we revealed that somata of these multi-dendritic neurons situated in the cibarium but not in the brain (*Figure 2G*). These multi-dendritic neurons in the cibarium may sense the swallowing process, and project to SEZ to interact with the neural circuits that control ingestion.

## md-C neurons are essential for swallow control

To explore the role of these pharyngeal multi-dendritic neurons, we expressed TNT (tetanus toxin) (*Martin et al., 2002*) in *Tmc* positive neurons to block the synaptic transmission, and found that flies' swallowing became arrhythmic and the pump frequency decreased significantly, with about 48.1% (n=26) flies displayed difficulty in cibarium emptying (*Video 3*; *Figure 2A and D*).

*Tmc* was also reported to participate in food texture sensation and proprioceptive control (*Zhang et al., 2016*; *He et al., 2019*; *Guo et al., 2016*; *Yue et al., 2019*), to restrict the manipulation to a more specific cell population, we used the FLP-out system to express Kir2.1 in neurons expressing both TMC and NOMPC. The intersection exclusively labels about two pairs of neurons around the

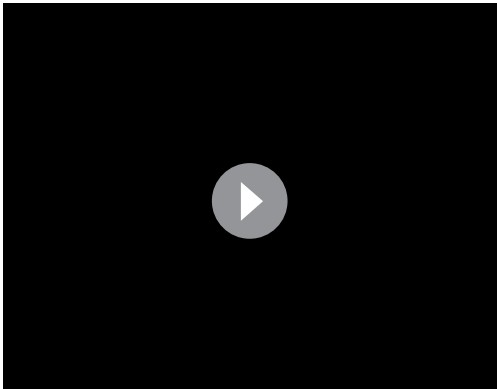

**Video 1.** Feeding behavior of wild-type flies and mechanoreceptor mutant flies.

https://elifesciences.org/articles/88614/figures#video1

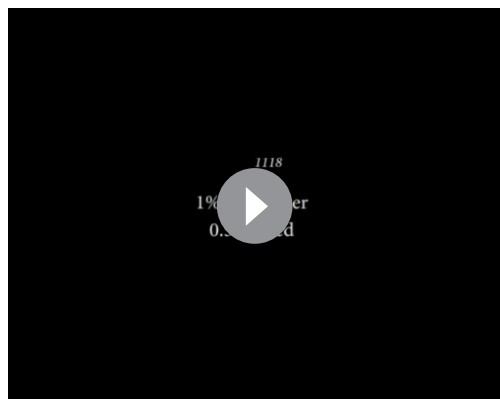

**Video 2.** *Tmc*[1] and *piezo*[KO] flies displayed incomplete emptying when fed with 1% methylcellulose (MC) water.

https://elifesciences.org/articles/88614/figures#video2

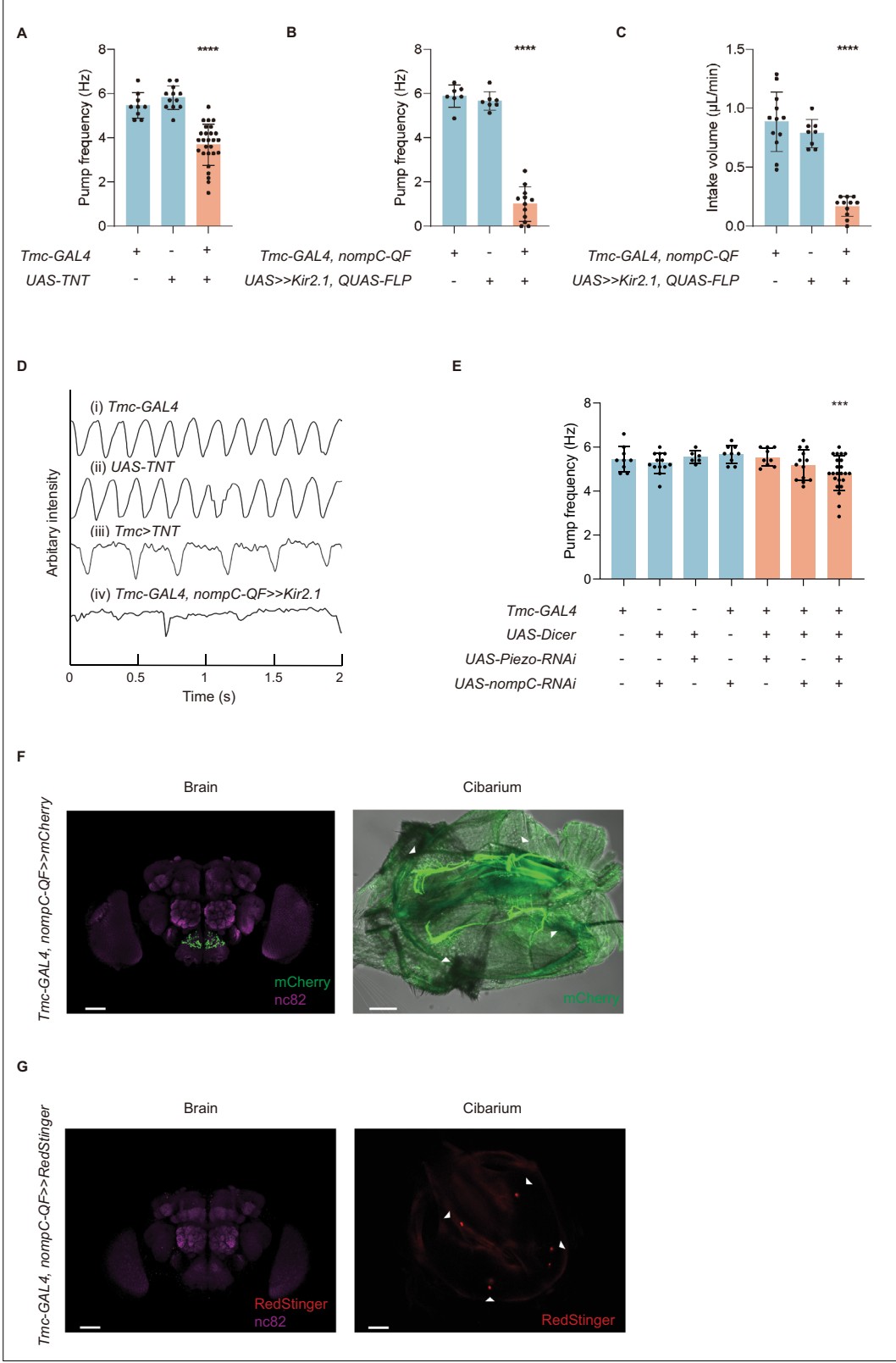

**Figure 2.** Mechanosensory neurons and mechanoreceptors are essential to swallow. (**A**) Blocking synaptic transduction of *Tmc* positive neurons leads to a lower pump frequency, and the flies display difficulty in cibarium emptying (emptying time>0.3 s). n=9–27 in each group. (**B–C**) Inhibiting *Tmc-GAL4* and *nompC-QF* double-labeled neurons by Kir2.1 results in a lower pump frequency and intake volume per minute. n=7–12 in each group.

*Figure 2 continued on next page*

*Figure 2 continued*

(**D**) Arbitrary intensity of cibarium when flies of different genotype swallow. The ordinate value was graphed using the opposite value of the original arbitrary intensity in the cibarium, so the declining line represents emptying process (the same for *Figure 3B*). (**E**) Pump frequency after knocking down the expression level of the mechanoreceptor genes. n=6–23 in each group. (**F**) Projection pattern of md-C neurons in the brain and cibarium. About two pairs of multi-dendritic neurons are situated along the pharynx. Genotype: *Tmc-GAL4; UAS-FRT-mCherry/nompC-QF; QUAS-FLP.* Scale bar=50 μm. (**G**) Somata of md-C neurons situated in the cibarium but not in the cibarium. Genotype: $X^{UAS-RedStinger}$; *Tmc-GAL4, UAS-FRT-GAL80-STOP-FRT/ +; nompC-QF, QUAS-FLP/+.* Scale bar=50 μm. In all analyses, one-way analysis of variance (ANOVA) followed by Dunnett's test for multiple comparisons was used, and statistical differences were represented as follows: *p<0.05, **p<0.01, ***p<0.001, and ****p<0.0001. Data were represented as means ± SEM.

The online version of this article includes the following figure supplement(s) for figure 2:

**Figure supplement 1.** Expression pattern of (**A**) *Tmc*, (**B**) *nompC*, (**C**) *piezo*, (**D**) *Tmc & piezo*, and (**E**) *nompC & piezo* in the brain and cibarium.

---

cibarium (*Figure 2F*) and they were named as md-C (multi-dendritic cibarium) neurons for brevity. We found that flies with md-C neurons inhibited displayed severe dysphagia (difficulty in swallowing), during which flies could hardly ingest water from the cibarium into the foregut. In some cases, water in the cibarium seemed to be pumped along the esophagus. However, no water was visible in the flies' crop, indicating that water in the cibarium was not successfully emptied (*Video 3*). As a result, flies' intake volume decreased dramatically (*Figure 2C*), indicating that md-C neurons are necessary for rhythmic pump and food ingestion.

We performed a double RNAi experiment for *nompC* and *Piezo* using *Tmc-GAL4* as a driver as the double mutant of these genes is lethal. Knocking down the expression level of *nompC* and *Piezo* resulted in a significantly lower pump frequency, similar to the frequency observed in flies with knockdown of either *nompC* or *Piezo* (*Figure 2E*). These results suggest that the channels may function complementarily.

While inhibiting md-C neurons caused difficulty in cibarium emptying, activating them optogenetically resulted in difficulty in cibarium filling. We activated md-C neurons with CsChrimson (*Klapoetke et al., 2014*) or ReaChR (*Lin et al., 2013*). In some cases, no dye could be seen in the cibarium although flies strongly extended their proboscis against water (*Figure 3A*, *Video 4*). This phenomenon could last for seconds but after several trials, flies can usually recover. In other cases, flies with md-C neurons activated show incomplete filling, indicated by decreased cibarium expansion, which was usually not observed in control groups (*Figure 3B*). Additionally, when flies consistently pump at a normal range (observed dye boundary in the cibarium is wider than 90% width of mouthpart), their pump frequency increased significantly (*Figure 3C*). Flies with md-C neurons activated showed a reduced volume per pump (*Figure 3C*), consistent with the small range of pump caused by incomplete filling. We thus speculated that the activation of

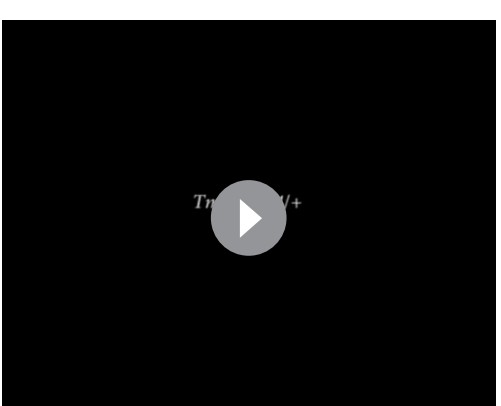

**Video 3.** Flies with md-C neurons inhibited displayed difficulty in cibarium emptying.
https://elifesciences.org/articles/88614/figures#video3

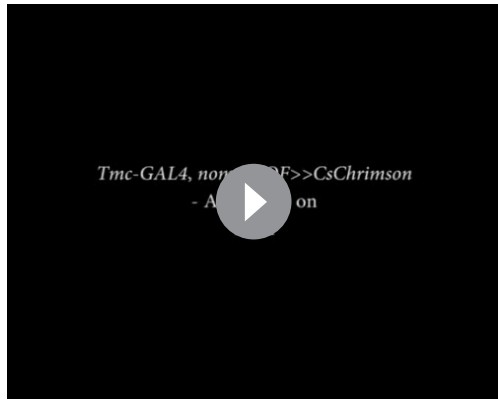

**Video 4.** Optogenetic activation of md-C neurons caused accelerated swallowing, incomplete cibarium filling, and difficulty in filling.
https://elifesciences.org/articles/88614/figures#video4

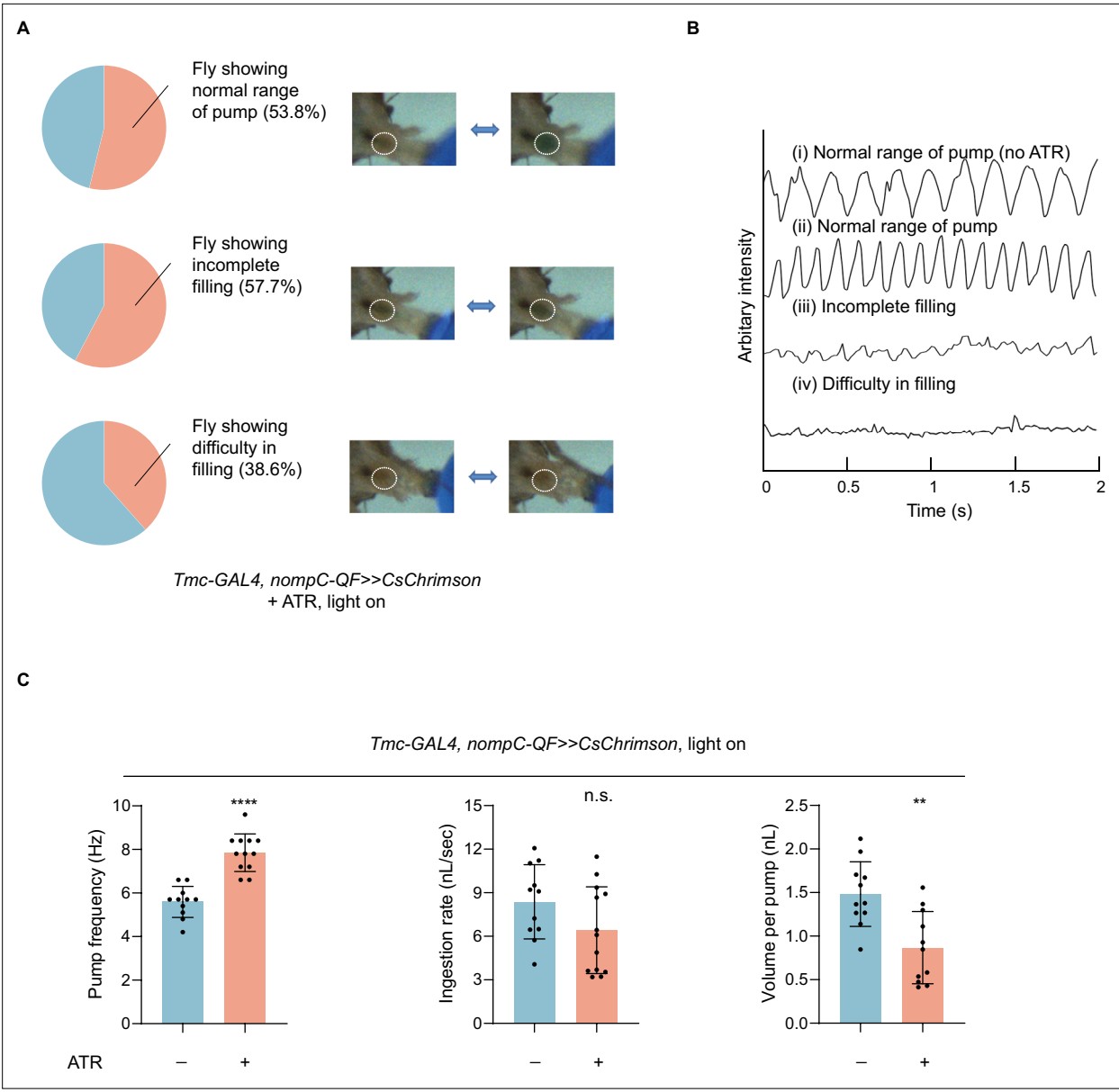

**Figure 3.** Activation of md-C neurons led to dysphagia. (**A**) Optogenetic activation of md-C neurons with CsChrimson during food intake could induce accelerated swallowing, incomplete filling (the expansion range of cibarium decreased to about half the maximum), and difficulty in filling. Distinct states driven by CsChrimson light stimulation of md-C neurons were displayed, with the proportions of flies exhibiting each state. n=26. (**B**) Arbitrary intensity of fly cibarium when md-C neurons were stimulated as (**A**) or not optogenetically stimulated (no ATR). (**C**) Swallowing behavior of flies with or without md-C neurons stimulated. n=11–13 in each group. Pump frequency was calculated only when fly consistently pumped at a normal range (observed dye boundary in the cibarium is wider than 90% width of mouthpart). ATR, all trans retinal. In all analyses, two-tailed unpaired t-tests were used, and statistical differences were represented as follows: **p<0.01 and ****p<0.0001. Data were represented as means ± SEM.

The online version of this article includes the following figure supplement(s) for figure 3:

**Figure supplement 1.** md-C neurons, but not md-L neurons, are essential for sensorimotor control of swallow rhythm.

md-C neurons could accelerate the emptying process but inhibit the filling process, which is opposite to their inhibition.

md-L neurons in the labellum (*Zhang et al., 2016*) could also be labeled by the intersection of *Tmc-GAL4* and *nompC-QF* (*Figure 3—figure supplement 1A*). To test whether the activation of md-C neurons alone was sufficient for the inhibition of cibarium filling, we cut the flies' labellum to ablate md-L neurons. We found that 36 hr after labellum ablation, the signals of md-L neurons could no longer be observed in the mouth (*Figure 3—figure supplement 1A*), indicating that the axons of md-L likely

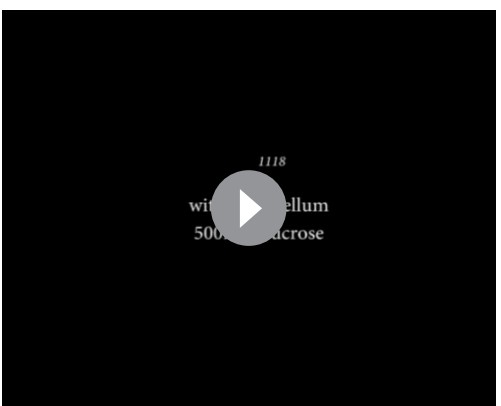

**Video 5.** Optogenetic activation of md-C neurons could still trigger dysphagia in flies without labellum. https://elifesciences.org/articles/88614/figures#video5

decomposed after 36 hr, rendering them ineffective in influencing swallowing. Besides, we found that after labellum ablation for 36 hr, light stimulation of md-C neurons still triggered filling difficulty despite the absence of md-L neurons, while these flies could pump at a normal range in the dark (*Figure 3—figure supplement 1B and C*, *Video 5*). We thus reasoned that md-C neurons in the cibarium, but not md-L neurons, were responsible for mechanical sensation during swallowing. We stained the muscles in the cibarium with phalloidin and labeled md-C neurons with GFP and found that both md-C neurons and muscles were in close proximity around cibarium (*Figure 4A*), suggesting that md-C neurons could be activated by muscle stretch during the expansion of cibarium.

Next, we wondered whether md-C neurons were activated by the action of swallowing. By expressing GCaMP6m in md-C neurons and cutting a small window in the head capsule, we could detect the fluorescent change of the md-C neurons' projections in the brain when flies were swallowing sugar food. An increase of fluorescence intensity in the SEZ was observed when flies were ingesting food (*Figure 4J and K*). As food ingestion delivered both gustatory and mechanosensory information, we asked whether md-C neurons were sensitive to taste input. The whole fly head was put into saline and a small window was cut (*Chen et al., 2019*). When sucrose was added to the saline, no signal changes were detected (*Figure 4K*), suggesting that md-C neurons responded to mechanical force but not gustatory cues during food ingestion.

## md-C neurons form synapses with MNs

So far, we have shown that md-C neurons regulate swallowing probably by sensing the expansion of the cibarium. But how do they coordinate with the downstream neural circuits to control the swallow? Two groups of MNs, MN11 and MN12, have been reported to control the muscles that execute the swallow process (*Manzo et al., 2012*; *McKellar et al., 2020*). We thus asked whether md-C neurons relay information of the cibarium volume to the MNs to control the swallow. We first tested their potential synaptic connection with GRASP (GFP reconstitution across synaptic partners) (*Feinberg et al., 2008*; *Gordon and Scott, 2009*), expressing one part of GFP in md-C neurons and the other part in MN12 or MN11. Signals could be detected in the SEZ where the neural projections of the two group neurons overlapped (*Figure 4B and C*), suggesting that md-C neurons may directly synapse onto the motor control circuit in the SEZ. By expressing P2X$_2$ (*North, 2002*; *Fountain, 2013*), an ATP receptor in $Tmc^+$ neurons, we also found that calcium transients could be observed in MN12 and MN11 when activating md-C neurons, while the control group show no significant differences (*Figure 4D–I*). Considering possible synaptic connections between md-C neurons and MNs, we suppose activation of md-C triggered by food bolus, might directly stimulate MNs to accelerate swallowing. These results suggest that MNs associated with swallowing are activated by sensory input in the cibarium to coordinate food ingestion during swallowing.

## Discussion

Here, we find that a group of mechanosensory neurons in the pharynx respond to the mechanical force generated by the expansion of the cibarium during food ingestion. These neurons may form synaptic and functional connections with MNs in the brain, and constitute an essential part of the neural circuit that controls the swallow.

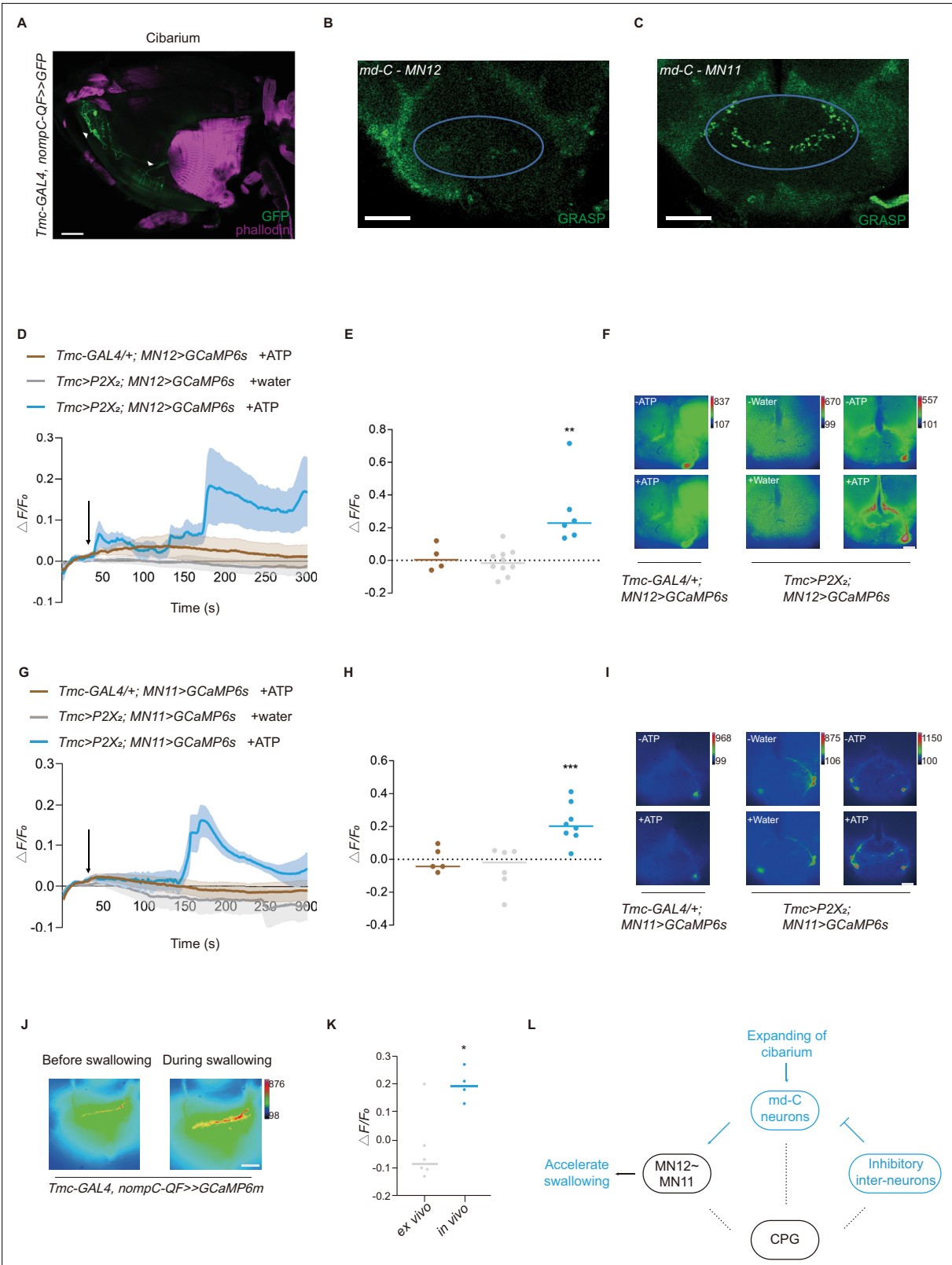

**Figure 4.** Interaction between motor neurons (MNs) and md-C neurons revealed working pattern of swallow. (**A**) md-C neurons labeled by GFP antibody (green) and muscles labeled by phalloidin (magenta) showed that they are in close proximity around cibarium. Genotype: *Tmc-GAL4/UAS-FRT-mCD8-GFP; nompC-QF/QUAS-FLP*. Scale bar=50 μm. (**B–C**) GRASP (GFP reconstitution across synaptic partners) signals between md-C neurons and MNs could be observed in the subesophageal zone (SEZ) area. Genotype: *Tmc-GAL4, UAS-FRT-GAL80-STOP-FRT/UAS-nSyb-spGFP1-10, lexAop-CD4-*

*Figure 4 continued*

*spGFP11; nompC-QF, QUAS-FLP/MN-LexA.* Scale bar=50 μm. (**D–F and G–I**) Activation of *Tmc*[+] neurons via P2X$_2$ increased MN activity. Fluorescence changes (ΔF/F$_0$) of GCaMP6s in MNs indicate calcium level changes. Water or ATP solution was added when it came to 30 s after record begins, as black arrow indicates. n=4–10, \*\*p<0.01, \*\*\*p<0.001, one-way analysis of variance (ANOVA) followed by Dunnett's test for multiple comparisons was used, error bars indicate mean ± SEM. Scale bar=50 μm. (**J**) Significant signal changes could be detected of the md-C neurons' termini at fly's SEZ area after feeding fly with 0.1 M sucrose solution when GCaMP6m is expressed in md-C neurons. Scale bar=50 μm. (**K**) are traces of fluorescence changes. n=4–6, \*p<0.05, two-tailed unpaired t-test was used, error bars indicate mean ± SEM. (**L**) Working model for md-C-MN-CPG controlling swallow of *Drosophila*. CPG, central pattern generators.

The online version of this article includes the following figure supplement(s) for figure 4:

**Figure supplement 1.** Regulation of md-C neurons requires the inhibitory inter-neurons.

## Multiple channels function in the same neurons

The mutants of either *nompC*, *Tmc*, or *piezo* showed defects in the swallow rhythm. However, we found that *nompC* seemed to play a more critical role as its mutation caused difficulty in emptying, while double mutation of *Tmc* and *piezo* did not. We speculated that *nompC* might be involved in swallowing liquid foods, as we used water in most experiments. While the md-C neurons also expressed both TMC and Piezo, we think these three channels could play different roles depending on physical properties of foods. The information generated by these three channels might be integrated in md-C neurons and helps the flies to distinguish the quality of food and regulate swallow.

## Interaction between md-C neurons and the feeding control circuit

It was demonstrated that MN12 controlled muscle 12 to fill cibarium with food, while MN11 controlled muscle 11 to expel the foods into the foregut (*Manzo et al., 2012*). In accordance with this model, we have found that the activation of md-C neurons could activate MN11, and over-stimulation of md-C neurons could cause difficulty in filling, incomplete filling, and/or higher pump frequency. On the other hand, silencing md-C neurons blocks the emptying process because expanding of cibarium fails to activate MN11 which controls emptying, leading to difficulty in swallowing. Additionally, when pharyngeal mechanosensation was impaired by the mutation of the mechanotransduction channel genes, MNs may not obtain enough activation to drive muscle 11 contraction to completely empty the foods in cibarium, resulting in a slower pump rhythm, especially when foods are of high viscosity.

However, the circuits that control the swallow could be more complex than the circuit proposed above. It has been reported that the ingestion of foods is completed in a sequential activity of MNs where most parts of the proboscis are engaged (*McKellar et al., 2020*). Besides the bottom-up interaction between md-C neurons and MN12/11 demonstrated in this study, a top-down feedback mechanism may also exist. For example, we have found that knocking down GABA$_A$-R in md-C neurons leads to a lower pump frequency (*Figure 4—figure supplement 1A*), suggesting that md-C neurons receive inhibitory signals during swallowing. Although it is unclear which neurons inhibit md-C neurons, we believe this is likely through a presynaptic mechanism on the axon termini of md-C neurons. Moreover, activating md-C induced calcium transients in MN12 and MN11 (*Figure 4D–I*), indicating that MNs were stimulated by activation of md-C neurons. It is believed that sequential contraction of muscles would cause fluid bolus move through the proboscis (*McKellar, 2016*), we thus hypothesize that md-C neurons might sequentially stimulate MNs during swallowing. Also, it is possible that sequential activation generated by MNs and md-C might crosstalk with CPGs, which together form a steady pump frequency, depending on the mechanical quality of foods (*Figure 4L*). Another possibility is that the activation of md-C neurons acts as a switch, altering the oscillation pattern of the swallowing CPGs from a resting state to a task state.

## CPGs in the control of swallow rhythm

It's widely accepted that most rhythmic behaviors of animals are controlled by the CPGs in the brain (*Marder and Bucher, 2007*; *Steuer and Guertin, 2019*). For example, brain stem CPGs control the patterns of suck, lick, mastication, swallow, etc. (*Marder and Bucher, 2007*; *Amirali et al., 2001*; *Grillner and El Manira, 2020*; *Yamaguchi et al., 2017*). Two lines of evidence support the notion that mechanical feedback from the cibarium during ingestion regulates the activity of CPGs or their downstream circuits that execute the pattern generated by CPGs. (1) High viscosity of food caused a reduction of pumping frequency while the rhythm remained essentially regular, indicating that the

pattern was modulated rather than blocked. (2) Manipulation of md-C neurons could interfere with the highly rhythmic pumping. Although the CPGs that control swallow in *Drosophila* brain are still elusive, our data provides an entry point to elucidate the CPGs circuits and the neural circuits associated with them. As their activity can be modulated by md-C neurons, they can be potentially identified by searching for the downstream of md-C neurons.

## Materials and methods
### Resource availability
#### Lead contact
Further information and requests for resources and reagents should be directed to and will be fulfilled by the lead contact, Wei Zhang, wei_zhang@mail.tsinghua.edu.cn.

#### Materials availability
This study did not generate new unique reagents. All key resources are listed in Key resources table. Further information and requests for resources and reagents should be directed to the lead contact.

### Experimental model and subject details
#### Animals
Fly were acquired from other labs or BDSC (Bloomington Drosophila Stock Center). Flies were reared on standard medium at 25°C unless otherwise noted. $w^{1118}$ was used as wild-type control. See Key resources table for the full list of fly strains.

#### Generation of transgenic flies
The *Tmc-GAL4^{DBD}*, *Piezo-GAL4^{AD}*, *MN12-lexA*, *MN11-LexA* line was constructed with the HACK system (*Lin and Potter, 2016*) from line *Tmc-GAL4*(BDSC:66557), *Piezo-GAL4*(BDSC:59266), *MN12-GAL4* and *MN11-GAL4* (donated from Kristin Scott lab). Phack plasmid was injected into nos-Cas9 flies with standard embryo injection procedures. Injected flies were crossed with double balancer flies, and the transformants were picked up with $w^+$ and RFP$^+$ makers. The *Tmc-GAL4,UAS(FRT.stop)CsChrimson-mVenus/CyO*, *nompC-QF,QUAS-FLP/TM6B* and *Tmc-GAL4,UAS-FRT-GAL80-STOP-FRT/CyO* lines were generated by recombination and confirmed by confocal imaging.

### Method details
#### Feeding assay
All flies were kept in a 24–26°C incubator under 12–12 hr light:dark cycle and about 40–60% humidity. Mated female flies were collected at eclosion and aged for 3–10 days. For neuronal silencing experiments, flies bearing Kir2.1 or TNT were kept for 7–10 days before the feeding assay. For rdl knockdown experiment in md-C neurons, 12- to 16-day-old flies were used. One day prior to the behavioral assays, flies were transferred to empty vials for 12-24 hours for water deprivation to keep them active and willing to drink.

Before experiments, flies were anesthetized with $CO_2$ and mounted onto a glass slide with nail polish. They were then allowed to recover in a humid chamber for 2 hr. All behavioral experiments were carried out at room temperature and 40–60% humidity. Individual flies were fed with a pipette filled with water, sugar, or MC water solutions. Consumption of single flies was calculated by measurable glass capillary. Ingestion rate was calculated by consumption divided by ingestion time, feeding lasted for 4–7 s except the experiment where Kir2.1 was used lasted for 1 min. For illustration, 0.25 mg/mL brilliant blue food dye (Shi-tou, GB 7655.1) was added to the solution. 60 fps camera was adopted to record the feeding behavior and flies with total feeding duration of more than 2 s were analyzed.

For optogenetics experiments, female flies expressing CsChrimson or ReaChR were raised on food containing 10 mM all-trans retinal at 25°C and 40–60% humidity in a light-proof vial. Experiments were performed with 7- to 10-day-old mated female flies in a dark room. One day before the test, flies were transferred into empty light-proof vials for water deprivation. Before and during feeding, flies were stimulated by 1.3 mW/cm$^2$ 590 nm light.

## Immunostaining and microscopy

The brains and cibaria of 3- to 10-day-old female flies were dissected in PBST dissection buffer containing 0.015% Triton X-100 in 1× PBS, followed by fixation in 4% PFA solution for 30 min on a shaker at room temperature. Samples were then washed four times for 20 min each with wash buffer (0.3% Triton X-100 in 1× PBS). The tissues were transferred to block buffer (1× heat-inactivated normal goat serum with 0.3% Triton X-100 in 1× PBS), and incubated at room temperature for 30 min. Primary antibodies were added to the samples and incubated overnight at 4°C. The primary antibodies used included Rabbit-RFP (Rockland 39707, diluted 1:500), Rabbit-GFP (Invitrogen A11122, diluted 1:500), Mouse-nc82 (Hybridoma Band DSHB, Brunchpilot, diluted 1:500), and Mouse-GFP (Sigma-Aldrich G6539, diluted 1:200). Samples were washed four times for 20 min each, and then incubated with secondary antibodies on the following day. The secondary antibodies were used at a 1:200 dilution and were all from Invitrogen: Alexa Fluor 555 anti-Rabbit (A-21428), Alexa Fluor 488 anti-Mouse (A11001), and Alexa Fluor 647 anti-Mouse (A-21235). After incubation for 5 hr at room temperature or overnight at 4°C, tissues were washed four times for 20 min each. The brains and cibaria were attached to a slide for imaging. Confocal imaging was performed using an Olympus FV1000 microscope with a 20× air lens.

## Functional imaging

Calcium imaging was carried out with an Olympus BX51WI microscope with 40× water immersion objective, Andor Zyla camera, and Uniblitz shutter. To perform calcium imaging in md-C neurons during swallowing, flies were food-deprived for 12 hr and heads were fixed in the place using nail polish, the antennae and the cuticle above the SEZ were carefully removed and the exposed brain was bathed in AHL buffer. Brains were dissected in a recording chamber containing saline (NaCl 140 mM, KCl 2 mM, $MgCl_2$ 4.5 mM, $CaCl_2$ 1.5 mM, HEPES-NaOH 5 mM, PH 7.1), and 1.5 mM $CaCl_2$ was added to the saline before use. ROIs and fluorescence changes were selected with ImageJ as peak $\Delta F/F_0$ = $(F_{peak} - F_0)/F_0$, where $F_0$ was the average fluorescence of 10 images when expressing GCaMP6m in md-C neurons before manipulation. Ex vivo control was performed as previously described (*Chen et al., 2019*).

While for GCaMP6s expressed in MNs, we immerse the fly in the imaging buffer. Following dissection and identification of the SEZ area under fluorescent microscopy, we introduce 20 mL water or 20 mL water solution of 250 mM ATP slowly into the liquid level, positioned at a distance from the brain, to minimize any potential interference with the procedure. Importantly, we did not administer ATP directly to the living fly. $F_0$ was the average fluorescence of 30 images in MNs before adding water or ATP to dishes containing 2 mL AHL buffer.

## Quantification and statistical analysis

Statistical analysis was performed with the GraphPad Prism 8 software. All error bars represent ± SEM. Two-tailed unpaired t-test and two-tailed Mann-Whitney nonparametric test were used to evaluate the significance between two datasets. For all analyses, statistical notations are as follows: *, $p < 0.05$. **, $p < 0.01$, ***, $p < 0.001$, ****, $p < 0.0001$. Number of dots per bar indicates the number of tested flies (N) in each experiment. No sample size estimation and inclusion and exclusion of any data or subjects were conducted in this study.

## Acknowledgements

We thank members of the Zhang lab for discussions. This work was supported by grants from the Innovation 2030 Major Project of the Ministry of Science and Technology of China (2021ZD0203300). This work was supported by grants 31871059 and 32022029 from the National Natural Science Foundation of China. This work is supported by Chinese Institute for Brain Research, Beijing. WZ is an awardee of the Young Thousand Talent Program of China.

## Additional information

### Funding

| Funder | Grant reference number | Author |
|---|---|---|
| National Natural Science Foundation of China | 31871059 | Wei Zhang |
| National Natural Science Foundation of China | 32022029 | Wei Zhang |
| Ministry of Science and Technology of China | 2021ZD0203300 | Wei Zhang |

The funders had no role in study design, data collection and interpretation, or the decision to submit the work for publication.

### Author contributions

Jierui Qin, Conceptualization, Resources, Data curation, Software, Formal analysis, Investigation, Visualization, Methodology, Writing – original draft, Writing – review and editing; Tingting Yang, Conceptualization, Resources, Data curation, Software, Formal analysis, Investigation, Visualization, Methodology; Kexin Li, Data curation, Investigation; Ting Liu, Resources, Data curation; Wei Zhang, Conceptualization, Resources, Data curation, Software, Formal analysis, Supervision, Funding acquisition, Validation, Investigation, Visualization, Methodology, Writing – original draft, Project administration, Writing – review and editing

### Author ORCIDs

Jierui Qin https://orcid.org/0009-0006-8197-1715
Wei Zhang https://orcid.org/0000-0003-0512-3096

Reviewer #1 (Public Review): https://doi.org/10.7554/eLife.88614.3.sa1
Reviewer #3 (Public Review): https://doi.org/10.7554/eLife.88614.3.sa2
Author response https://doi.org/10.7554/eLife.88614.3.sa3

## Additional files

### Supplementary files

• MDAR checklist

### Data availability

All data generated or analysed during this study are included in the manuscript and the dataset uploaded in Dryad (https://doi.org/10.5061/dryad.vdncjsz4q).

The following dataset was generated:

| Author(s) | Year | Dataset title | Dataset URL | Database and Identifier |
|---|---|---|---|---|
| Zhang W, Qin J, Yang T, Li K, Liu T | 2024 | Data From: Pharyngeal Mechanosensory Neurons Control Food Swallow in *Drosophila melanogaster* | https://doi.org/10.5061/dryad.vdncjsz4q | Dryad Digital Repository, 10.5061/dryad.vdncjsz4q |

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

# Appendix 1

## Appendix 1—key resources table

| Reagent type (species) or resource | Designation | Source or reference | Identifiers | Additional information |
|---|---|---|---|---|
| Genetic reagent (*D. melanogaster*) | Tmc-Gal4 | Bloomington *Drosophila* Stock Center | BDSC:66557 | |
| Genetic reagent (*D. melanogaster*) | QUAS-FLP | Bloomington *Drosophila* Stock Center | BDSC:30127 | |
| Genetic reagent (*D. melanogaster*) | UAS-FRT-GAL80-STOP-FRT | Bloomington *Drosophila* Stock Center | BDSC:38880 | |
| Genetic reagent (*D. melanogaster*) | UAS-FRT-STOP-Kir2.1-FRT | Bloomington *Drosophila* Stock Center | BDSC:67686 | |
| Genetic reagent (*D. melanogaster*) | UAS(FRT.mCherry)ReaChR | Bloomington *Drosophila* Stock Center | BDSC:53743 | |
| Genetic reagent (*D. melanogaster*) | UAS-TNT | Bloomington *Drosophila* Stock Center | BDSC:28997 | |
| Genetic reagent (*D. melanogaster*) | LexAop-GCaMP6s | Bloomington *Drosophila* Stock Center | BDSC:64413 | |
| Genetic reagent (*D. melanogaster*) | Tmc[1] | Bloomington *Drosophila* Stock Center | BDSC:66556 | |
| Genetic reagent (*D. melanogaster*) | w[*]; P{w[+Mc] = UAS-nSyb-spGFP1-10}2, P{w[+Mc] = lexAop-CD4-spGFP11}2/CyO | Bloomington *Drosophila* Stock Center | BDSC:64314 | |
| Genetic reagent (*D. melanogaster*) | UAS-RedStinger | Bloomington *Drosophila* Stock Center | BDSC:8547 | |
| Genetic reagent (*D. melanogaster*) | Piezo-GAL4 | Bloomington *Drosophila* Stock Center | BDSC:59266 | |
| Genetic reagent (*D. melanogaster*) | UAS-Piezo-RNAi | Vienna *Drosophila* RNAi Center | VDRC:105132 | |
| Genetic reagent (*D. melanogaster*) | UAS-FRT-STOP-FRT-GCaMP6m | other | | From Laboratory of Chuan Zhou, Institute of Zoology, Chinese Academy of Sciences |
| Genetic reagent (*D. melanogaster*) | nompC-QF | *Yang et al., 2021* | | From Laboratory of Yuh Nung Jan, UCSF |
| Genetic reagent (*D. melanogaster*) | nompC[1]/CyO | *Yang et al., 2021* | | From Laboratory of Yuh Nung Jan, UCSF |
| Genetic reagent (*D. melanogaster*) | nompC[f00914]/CyO | *Yang et al., 2021* | | From Laboratory of Yuh Nung Jan, UCSF |
| Genetic reagent (*D. melanogaster*) | piezo[KO] | *Zhang et al., 2013* | | From Laboratory of Yuh Nung Jan, UCSF |
| Genetic reagent (*D. melanogaster*) | UAS-nompC-RNAi; UAS-Dicer2 | *Yang et al., 2021* | | From Laboratory of Yuh Nung Jan, UCSF |
| Genetic reagent (*D. melanogaster*) | UAS-rdl-RNAi | *Yang et al., 2021* | | From Laboratory of Xin Liang, THU |
| Genetic reagent (*D. melanogaster*) | UAS-P2X$_2$ | *Yang et al., 2021* | | From Laboratory of Yufeng Pan, School of Life Science and Technology, Southeast University |
| Genetic reagent (*D. melanogaster*) | UAS(FRT.stop)CsChrimson-mVenus | *Wu et al., 2019a* | | From Laboratory of Yufeng Pan, School of Life Science and Technology, Southeast University |
| Genetic reagent (*D. melanogaster*) | MN12-GAL4 | *Manzo et al., 2012* | | From Laboratory of Kristin Scott, UCB |
| Genetic reagent (*D. melanogaster*) | MN11-GAL4 | *Manzo et al., 2012* | | From Laboratory of Kristin Scott, UCB |
| Genetic reagent (*D. melanogaster*) | MN12-LexA | This paper | | Hacked from MN12-GAL4 |
| Genetic reagent (*D. melanogaster*) | MN11-LexA | This paper | | Hacked from MN11-GAL4 |

*Appendix 1 Continued on next page*

*Appendix 1 Continued*

| Reagent type (species) or resource | Designation | Source or reference | Identifiers | Additional information |
|---|---|---|---|---|
| Genetic reagent (*D. melanogaster*) | Tmc-GAL4, UAS(FRT.stop)CsChrimson-mVenus/ CyO | This paper | | Recombined from BDSC:66557 and UAS(FRT. stop)CsChrimson-mVenus |
| Genetic reagent (*D. melanogaster*) | nompC-QF, QUAS-Flp/TM6B | This paper | | Recombined from BDSC:30127 and nompC-QF |
| Genetic reagent (*D. melanogaster*) | Tmc-GAL4, UAS-FRT-GAL80-STOP-FRT/CyO | This paper | | Recombined from BDSC:66557&38880 |
| Genetic reagent (*D. melanogaster*) | Piezo-GAL4$^{AD}$ | This paper | | Hacked from BDSC:59266 |
| Genetic reagent (*D. melanogaster*) | Tmc-GAL4$^{DBD}$ | This paper | | Hacked from BDSC:66557 |
| Antibody | anti-Rabbit Alexa 488(Goat polyclonal) | Invitrogen | Cat#A11008; RRID: AB_143165 | 1:200 |
| Antibody | anti-Rabbit Alexa 555(Goat polyclonal) | Invitrogen | Cat#A21428; RRID: AB_2535849 | 1:200 |
| Antibody | anti-Mouse Alexa 488(Goat polyclonal) | Invitrogen | Cat#A11001; RRID: AB_2534069 | 1:200 |
| Antibody | anti-Mouse Alexa 647(Goat polyclonal) | Invitrogen | Cat#A21235; RRID: AB_2535804 | 1:200 |
| Antibody | anti-Brp(Mouse monoclonal) | Developmental Studies Hybridoma Bank | Cat# nc82; RRID: AB_2314866 | 1:500 |
| Antibody | anti-GFP(Rabbit polyclonal) | Invitrogen | Cat#A11122; RRID: AB_221569 | 1:500 |
| Antibody | anti-RFP(Rabbit polyclonal) | Rockland | Cat#600-401-379S; RRID: AB_11182807 | 1:500 |
| Antibody | anti-GFP(Mouse monoclonal) | Sigma-Aldrich | Cat# G6539; RRID: AB_259941 | 1:200 |
| Chemical compound, drug | 4% PFA | Dingguo Biotech | Cat#AR-0211;CAS:30525-89-4 | |
| Chemical compound, drug | Brilliant Blue | Shi-tou | Cat#GB 7655.1 | |
| Chemical compound, drug | Methylcellulose | Sigma-Aldrich | M7140-100G | |
| Chemical compound, drug | All trans retinal | Sigma-Aldrich | R2500-500MG | |
| Chemical compound, drug | Phalloidin | AAT bioquest | Cat#23127 | |
| Chemical compound, drug | Adenosine-5 | aladdin | Lot#I2118087 | |
| Software, algorithm | ImageJ and Fiji | NIH; *Schindelin et al., 2012* | https://imagej.nih.gov/ij/ http://fiji.sc/ | |
| Software, algorithm | GraphPad Prism 8 | GraphPad Software | https://www.graphpad.com/ scientific-software/prism/ | |
| Software, algorithm | Adobe Illustrator | Adobe Systems | https://www.adobe.com/ products/illustrator.html | |

