## [Editor Report · eLife Assessment]

This **valuable** study investigates the role of mechanosensory feedback during swallowing in adult *Drosophila*. The authors provide **convincing** evidence that three mechanotransduction channel genes are required for ingestion rhythms and localize the role of these genes to a specific subpopulation of pharyngeal mechanosensory neurons. However, there is incomplete evidence to support the conclusions that these sensory neurons are necessary for swallowing, respond to stretch during swallowing, and connect to the motor neurons that control swallowing. This work may be of interest to neuroscientists interested in motor control of feeding behavior.

---

## [Referee Report · Reviewer #1 (Public Review)]

Qin et al. set out to investigate the role of mechanosensory feedback during swallowing and identify neural circuits that generate ingestion rhythms. They use *Drosophila melanogaster* swallowing as a model system, focusing their study on the neural mechanisms that control cibarium filling and emptying in vivo. They find that pump frequency is decreased in mutants of three mechanotransduction genes (nompC, piezo, and Tmc), and conclude that mechanosensation mainly contributes to the emptying phase of swallowing. Furthermore, they find that double mutants of nompC and Tmc have more pronounced cibarium pumping defects than either single mutants or Tmc/piezo double mutants. They discovered that the expression patterns of nompC and Tmc overlap in two classes of neurons, md-C and md-L neurons. The dendrites of md-C neurons warp the cibarium and project their axons to the subesophageal zone of the brain. Silencing neurons that express both nompC and Tmc leads to severe ingestion defects, with decreased cibarium emptying. Optogenetic activation of the same population of neurons inhibited filling of the cibarium and accelerated cibarium emptying. In the brain, the axons of nompC∩Tmc cell types respond during ingestion of sugar but do not respond when the entire fly head is passively exposed to sucrose. Finally, the authors show that nompC∩Tmc cell types arborize close to the dendrites of motor neurons that are required for swallowing and that swallowing motor neurons respond to the activation of the entire Tmc-GAL4 pattern.

Strengths:

-The authors rigorously quantify ingestion behavior to convincingly demonstrate the importance of mechanosensory genes in the control of swallowing rhythms and cibarium filling and emptying

-The authors demonstrate that a small population of neurons that express both nompC and Tmc oppositely regulate cibarium emptying and filling when inhibited or activated, respectively

-They provide evidence that the action of multiple mechanotransduction genes may converge in common cell types

Weaknesses:

-A major weakness of the paper is that the authors use reagents that are expressed in both md-C and md-L but describe the results as though only md-C is manipulated

-Evidence that the defects they see in pumping can be specifically attributed to md-C is based on severing the labellum and allowing md-L neurons to degrade.

-GRASP is known to be non-specific and prone to false positives when neurons are in close proximity but not synaptically connected. A positive GRASP signal supports but does not confirm direct synaptic connectivity between md-C/md-L axons and MN11/MN12.

-MN11/MN12 LexA lines are not included in the manuscript and their expression patterns (shared with the reviewers in the author response) do not appear to contain any motor neurons. Double labeling with previously described MN11 and MN12 motor neuron Gal4 lines is needed to support the claim that these LexA lines in fact label MN11 and MN12.

-As seen in Figure Supplement 2, the expression pattern of Tmc-GAL4 is broader than md-C alone. Therefore, the functional connectivity the authors observe between Tmc expressing neurons and MN11 and 12 cannot be traced to md-C alone

-Example traces of md-C calcium imaging during ingestion in vivo are not included, and evidence that md-C neurons respond to mechanical force is lacking

-A positive control (perhaps demonstrating that sugar sensory neurons respond to sucrose in this preparation) is needed to assess whether the lack of response to sucrose ex vivo in Figure 4K is informative

-Proximity between md-C neurons and muscles is not evidence that they sense stretch

-Reporting of posthoc tests needs to be improved throughout the manuscript, as it is not clear which comparisons are noted with asterisks in the figures.

Overall, this work convincingly shows that swallowing and swallowing rhythms are dependent on several mechanosensory genes. Qin et al. also characterize a candidate neuron, md-C, that is likely to provide mechanosensory feedback to pumping motor neurons, but the results they present here are not sufficient to assign this function to md-C alone. This work will have a positive impact on the field by demonstrating the importance of mechanosensory feedback to swallowing rhythms and providing a potential entry point for future investigation of the identity and mechanisms of swallowing central pattern generators.

---

## [Referee Report · Reviewer #3 (Public Review)]

Strengths:

There are many reports on the effect of chemical properties of foods on feeding in fruit flies, but only limited studies reported how physical properties of food affect feeding especially pharyngeal mechanosensory neurons. First, they found that mechanosensory mutants, including nompC, Tmc, and Piezo, showed impaired swallowing, mainly the emptying process. Next, they identified cibarium multidendritic mechanosensory neurons (md-C) are responsible for controlling swallowing by regulating motor neuron (MN) 12 and 11, which control filling and emptying, respectively.

Weaknesses:

While the involvement of md-C and mechanosensory channels in controlling swallowing is convincing, it is not yet clear which stimuli activate md-C. Can it be an expansion of cibarium or food viscosity, or both? In addition, if rhythmic and coordinated contraction of muscles 11 and 12 is essential for swallowing, how can simultaneous activation of MN 11 and 12 by md-C achieve this? Finally, previous reports showed that food viscosity mainly affects the filling rather than the emptying process, which seems different from their finding.

---

## [Author Response]

The following is the authors’ response to the original reviews.

**Public Reviews:**

**Reviewer #1 (Public Review):**
Qin et al. set out to investigate the role of mechanosensory feedback during swallowing and identify neural circuits that generate ingestion rhythms. They use *Drosophila melanogaster* swallowing as a model system, focusing their study on the neural mechanisms that control cibarium filling and emptying in vivo. They find that pump frequency is decreased in mutants of three mechanotransduction genes (nompC, piezo, and Tmc), and conclude that mechanosensation mainly contributes to the emptying phase of swallowing. Furthermore, they find that double mutants of nompC and Tmc have more pronounced cibarium pumping defects than either single mutants or Tmc/piezo double mutants. They discover that the expression patterns of nompC and Tmc overlap in two classes of neurons, md-C and md-L neurons. The dendrites of md-C neurons warp the cibarium and project their axons to the subesophageal zone of the brain. Silencing neurons that express both nompC and Tmc leads to severe ingestion defects, with decreased cibarium emptying. Optogenetic activation of the same population of neurons inhibited filling of the cibarium and accelerated cibarium emptying. In the brain, the axons of nompC∩Tmc cell types respond during ingestion of sugar but do not respond when the entire fly head is passively exposed to sucrose. Finally, the authors show that nompC∩Tmc cell types arborize close to the dendrites of motor neurons that are required for swallowing, and that swallowing motor neurons respond to the activation of the entire Tmc-GAL4 pattern.Strengths:The authors rigorously quantify ingestion behavior to convincingly demonstrate the importance of mechanosensory genes in the control of swallowing rhythms and cibarium filling and emptyingThe authors demonstrate that a small population of neurons that express both nompC and Tmc oppositely regulate cibarium emptying and filling when inhibited or activated, respectivelyThey provide evidence that the action of multiple mechanotransduction genes may converge in common cell types

Thank you for your insightful and detailed assessment of our work. Your constructive feedback will help to improve our manuscript.

Weaknesses:A major weakness of the paper is that the authors use reagents that are expressed in both md-C and md-L but describe the results as though only md-C is manipulated-Severing the labellum will not prevent optogenetic activation of md-L from triggering neural responses downstream of md-L. Optogenetic activation is strong enough to trigger action potentials in the remaining axons. Therefore, Qin et al. do not present convincing evidence that the defects they see in pumping can be specifically attributed to md-C.

Thank you for your comments. This is important point that we did not adequately address in the original preprint. We have obtained imaging and behavioral results that strongly suggest md-C, rather than md-L, are essential for swallowing behavior.

36 hours after the ablation of the labellum, the signals of md-L were hardly observable when GFP expression was driven by the intersection between Tmc-GAL4 & nompC-QF (see F Figure 3—figure supplement 1A). This observation indicates that the axons of md-L likely degenerated after 36 hours, and were unlikely to influence swallowing. Moreover, the projecting pattern of Tmc-GAL4 & nompC-QF>>GFP exhibited no significant changes in the brain post labellum ablation.

Furthermore, even after labellum ablation for 36 hours, flies exhibited responses to light stimulation (see Figure 3—figure supplement 1B-C, Video 5) when ReaChR was expressed in md-C. We thus reasoned that md-C but not md-L, plays a crucial role in the swallowing process.

GRASP is known to be non-specific and prone to false positives when neurons are in close proximity but not synaptically connected. A positive GRASP signal supports but does not confirm direct synaptic connectivity between md-C/md-L axons and MN11/MN12.

In this study, we employed the nSyb-GRASP, wherein the GRASP is expressed at the presynaptic terminals by fusion with the synaptic marker nSyb. This method demonstrates an enhanced specificity compared to the original GRASP approach.

Additionally, we utilized +/ UAS-nSyb-spGFP1-10, lexAop-CD4-spGFP11 ; + / MN-LexA fruit flies as a negative control to mitigate potential false signals originating from the tool itself (Author response image 1, scale bar = 50μm). Beside the genotype Tmc-Gal4, Tub(FRT. Gal80) / UAS-nSyb-spGFP1-10, lexAop-CD4-spGFP11 ; nompC-QF, QUAS-FLP / MN-LexA fruit flies discussed in this manuscript, we also incorporated genotype Tmc-Gal4, Tub(FRT. Gal80) / lexAop-nSyb-spGFP1-10, UAS-CD4-spGFP11 ; nompC-QF, QUAS-FLP / MN-LexA fruit flies as a reverse control (Author response image 2). Unexpectedly, similar positive signals were observed, indicating that, positive signals may emerge due to close proximity between neurons even with nSyb-GRASP.

Author response image 1

It should be noted that the existence of synaptic projections from motor neurons (MN) to md-C cannot be definitively confirmed at this juncture. At present, we can only posit the potential for synaptic connections between md-C and motor neurons. A more conclusive conclusion may be attainable with the utilization of comprehensive whole-brain connectome data in future studies.

Author response image 2

As seen in Figure 2—figure supplement 1, the expression pattern of Tmc-GAL4 is broader than md-C alone. Therefore, the functional connectivity the authors observe between Tmc expressing neurons and MN11 and 12 cannot be traced to md-C alone

It is true that the expression pattern of Tmc-GAL4 is broader than that of md-C alone. Our experiments, including those flies expressing TNT in Tmc+ neurons, demonstrated difficulties in emptying (Figure 2A, 2D). Notably, we encountered challenges in finding fly stocks bearing UAS>FRT-STOP-P2X2. Consequently, we opted to utilize Tmc-GAL4 to drive UAS-P2X2 instead. We believe that the results further support our hypothesis on the role of md-C in the observed behavioral change in emptying.

Overall, this work convincingly shows that swallowing and swallowing rhythms are dependent on several mechanosensory genes. Qin et al. also characterize a candidate neuron, md-C, that is likely to provide mechanosensory feedback to pumping motor neurons, but the results they present here are not sufficient to assign this function to md-C alone. This work will have a positive impact on the field by demonstrating the importance of mechanosensory feedback to swallowing rhythms and providing a potential entry point for future investigation of the identity and mechanisms of swallowing central pattern generators.
**Reviewer #2 (Public Review):**
In this manuscript, the authors describe the role of cibarial mechanosensory neurons in fly ingestion. They demonstrate that pumping of the cibarium is subtly disrupted in mutants for piezo, TMC, and nomp-C. Evidence is presented that these three genes are co-expressed in a set of cibarial mechanosensory neurons named md-C. Silencing of md-C neurons results in disrupted cibarial emptying, while activation promotes faster pumping and/or difficulty filling. GRASP and chemogenetic activation of the md-C neurons is used to argue that they may be directly connected to motor neurons that control cibarial emptying.The manuscript makes several convincing and useful contributions. First, identifying the md-C neurons and demonstrating their essential role for cibarium emptying provides reagents for further studying this circuit and also demonstrates the important of mechanosensation in driving pumping rhythms in the pharynx. Second, the suggestion that these mechanosensory neurons are directly connected to motor neurons controlling pumping stands in contrast to other sensory circuits identified in fly feeding and is an interesting idea that can be more rigorously tested in the future.At the same time, there are several shortcomings that limit the scope of the paper and the confidence in some claims. These include:a) the MN-LexA lines used for GRASP experiments are not characterized in any other way to demonstrate specificity. These were generated for this study using Phack methods, and their expression should be shown to be specific for MN11 and MN12 in order to interpret the GRASP experiments.

Thanks for the suggestion. We have checked the expression pattern of MN-LexA, which is similar to MN-GAL4 used in previous work (Manzo et al., PNAS., 2012, PMID:22474379) . Here is the expression pattern:

Author response image 3

9%

b) There is also insufficient detail for the P2X2 experiment to evaluate its results. Is this an in vivo or ex vivo prep? Is ATP added to the brain, or ingested? If it is ingested, how is ATP coming into contact with md-C neuron if it is not a chemosensory neuron and therefore not exposed to the contents of the cibarium?

The P2X2 experimental preparation was done ex vivo. We immersed the fly in the imaging buffer, as described in the Methods section under Functional Imaging. Following dissection and identification of the subesophageal zone (SEZ) area under fluorescent microscopy, we introduced ATP slowly into the buffer, positioned at a distance from the brain

c) In Figure 3C, the authors claim that ablating the labellum will remove the optogenetic stimulation of the md-L neuron (mechanosensory neuron of the labellum), but this manipulation would presumably leave an intact md-L axon that would still be capable of being optogenetically activated by Chrimson.

Please refer to the corresponding answers for reviewer 1 and Figure 3—figure supplement 1.

d) Average GCaMP traces are not shown for md-C during ingestion, and therefore it is impossible to gauge the dynamics of md-C neuron activation during swallowing. Seeing activation with a similar frequency to pumping would support the suggested role for these neurons, although GCaMP6s may be too slow for these purposes.

Profiling the dynamics of md-C neuron activation during swallowing is crucial for unraveling the operational model of md-C and validating our proposed hypothesis. Unfortunately, our assay faces challenges in detecting probable 6Hz fluorescent changes with GCaMP6s.

In general, we observed an increase of fluorescent signals during swallowing, but movement of alive flies during swallowing influenced the imaging recording, so we could not depict a decent tracing for calcium imaging for md-C neurons. To enhance the robustness of our findings, patching the md-C neurons would be a more convincing approach. As illustrated in Figure 2, the somata of md-C neurons are situated in the cibarium rather than the brain. patching of the md-C neuron somata in flies during ingestion is difficult.

e) The negative result in Figure 4K that is meant to rule out taste stimulation of md-C is not useful without a positive control for pharyngeal taste neuron activation in this same preparation.

We followed methods used in the previous work (Chen et al., Cell Rep., 2019, PMID:31644916), which we believe could confirm that md-C do not respond to sugars.

In addition to the experimental limitations described above, the manuscript could be organized in a way that is easier to read (for example, not jumping back and forth in figure order).

Thanks for your suggestion and the manuscript has been reorganized.

**Reviewer #3 (Public Review):**
Swallowing is an essential daily activity for survival, and pharyngo-laryngeal sensory function is critical for safe swallowing. In *Drosophila*, it has been reported that the mechanical property of food (e.g. Viscosity) can modulate swallowing. However, how mechanical expansion of the pharynx or fluid content sense and control swallowing was elusive. Qin et al. showed that a group of pharyngeal mechanosensory neurons, as well as mechanosensory channels (nompC, Tmc, and Piezo), respond to these mechanical forces for regulation of swallowing in *Drosophila melanogaster*.Strengths:There are many reports on the effect of chemical properties of foods on feeding in fruit flies, but only limited studies reported how physical properties of food affect feeding especially pharyngeal mechanosensory neurons. First, they found that mechanosensory mutants, including nompC, Tmc, and Piezo, showed impaired swallowing, mainly the emptying process. Next, they identified cibarium multidendritic mechanosensory neurons (md-C) are responsible for controlling swallowing by regulating motor neuron (MN) 12 and 11, which control filling and emptying, respectively.Weaknesses:While the involvement of md-C and mechanosensory channels in controlling swallowing is convincing, it is not yet clear which stimuli activate md-C. Can it be an expansion of cibarium or food viscosity, or both? In addition, if rhythmic and coordinated contraction of muscles 11 and 12 is essential for swallowing, how can simultaneous activation of MN 11 and 12 by md-C achieve this? Finally, previous reports showed that food viscosity mainly affects the filling rather than the emptying process, which seems different from their finding.

We have confirmed that swallowing sucrose water solution activated md-C neurons, while sucrose water solution alone could not (Figure 4J-K). We hypothesized that the viscosity of the food might influence this expansion process.

While we were unable to delineate the activation dynamics of md-C neurons, our proposal posits that these neurons could be activated in a single pump cycle, sequentially stimulating MN12 and MN11. Another possibility is that the activation of md-C neurons acts as a switch, altering the oscillation pattern of the swallowing central pattern generator (CPG) from a resting state to a working state.

In the experiments with w1118 flies fed with MC (methylcellulose) water, we observed that viscosity predominantly affects the filling process rather than the emptying process, consistent with previous findings. This raises an intriguing question. Our investigation into the mutation of mechanosensitive ion channels revealed a significant impact on the emptying process. We believe this is due to the loss of mechanosensation affecting the vibration of swallowing circuits, thereby influencing both the emptying and filling processes. In contrast, viscosity appears to make it more challenging for the fly to fill the cibarium with food, primarily attributable to the inherent properties of the food itself.

**Reviewer #4 (Public Review):**
A combination of optogenetic behavioral experiments and functional imaging are employed to identify the role of mechanosensory neurons in food swallowing in adult *Drosophila*. While some of the findings are intriguing and the overall goal of mapping a sensory to motor circuit for this rhythmic movement are admirable, the data presented could be improved.The circuit proposed (and supported by GRASP contact data) shows these multi-dendritic neurons connecting to pharyngeal motor neurons. This is pretty direct - there is no evidence that they affect the hypothetical central pattern generator - just the execution of its rhythm. The optogenetic activation and inhibition experiments are constitutive, not patterned light, and they seem to disrupt the timing of pumping, not impose a new one. A slight slowing of the rhythm is not consistent with the proposed function.

Motor neurons implicated in patterned motions can be considered effectors of Central Pattern Generators (CPGs)(Marder et al., Curr Biol., 2001, PMID: 11728329; Hurkey et al., Nature., 2023, PMID:37225999). Given our observation of the connection between md-C neurons and motor neurons, it is reasonable to speculate that md-C neurons influence CPGs. Compared to the patterned light (0.1s light on and 0.1s light off) used in our optogenetic experiments, we noted no significant changes in their responses to continuous light stimulation. We think that optogenetic methods may lead to overstimulation of md-C neurons, failing to accurately mimic the expansion of the cibarium during feeding.

Dysfunction in mechanosensitive ion channels or mechanosensory neurons not only disrupts the timing of pumping but also results in decreased intake efficiency (Figure 1E). The water-swallowing rhythm is generally stable in flies, and swallowing is a vital process that may involve redundant ion channels to ensure its stability.

The mechanosensory channel mutants nompC, piezo, and TMC have a range of defects. The role of these channels in swallowing may not be sufficiently specific to support the interpretation presented. Their other defects are not described here and their overall locomotor function is not measured. If the flies have trouble consuming sufficient food throughout their development, how healthy are they at the time of assay? The level of starvation or water deprivation can affect different properties of feeding - meal size and frequency. There is no description of how starvation state was standardized or measured in these experiments.

Defects in mechanosensory channel mutants nompC, piezo, and TMC, have been extensively investigated (Hehlert et al., Trends Neurosci., 2021, PMID:332570000). Mutations in these channels exhibit multifaceted effects, as illustrated in our RNAi experiments (see Figure 2E).Deprivation of water and food was performed in empty fly vials. It's important to note that the duration of starvation determines the fly's willingness to feed but not the pump frequency (Manzo et al., PNAS., 2012, PMID:22474379).

In most cases, female flies were deprived water and food in empty vials for 24 hours because after that most flies would be willing to drink water. The deprivation time is 12 hours for flies with nompC and Tmc mutated or flies with Kir2.1 expressed in md-C neurons, as some of these flies cannot survive 24h deprivation.

The brain is likely to move considerably during swallow, so the GCaMP signal change may be a motion artifact. Sometimes this can be calculated by comparing GCaMP signal to that of a co-expressed fluorescent protein, but there is no mention that this is done here. Therefore, the GCaMP data cannot be interpreted.

We did not co-express a fluorescent protein with GCaMP for md-C. The head of the fly was mounted onto a glass slide, and we did not observe significant signal changes before feeding.

**Recommendations for the authors:**

**Reviewer #1 (Recommendations For The Authors):**

.>Abstract: I disagree that swallow is the first step of ingestion. The first paragraph also mentions the final checkpoint before food ingestion. Perhaps sufficient to say that swallow is a critical step of ingestion.

Indeed, it is not rigorous enough to say “first step”. This has been replaced by “early step”.

Introduction:Line 59: "Silence" should be "Silencing"This has been replaced.Results:Lines 91-92: I am not clear about what this means. 20% of nompC and 20% of wild-type flies exhibit incomplete filling? So nompC is not different from wild-type?

Sorry for the mistake. Viscous foods led to incomplete emptying (not incomplete filling), as displayed in Video 4. The swallowing behavior differs between nompC mutants and wild-type flies, as illustrated in Figure 1C, Figure 1—figure supplement 1A-C and video 1&5.

When fed with 1% MC water solution (Figure 1—figure supplement 1E-H). We found that when fed with 1% MC watere solution, Tmc or piezo mutants displayed incomplete emptying, which could constitute a long time proportion of swallowing behavior; while only 20% of nompC flies and 20% of wild-type flies sporadically exhibit incomplete emptying, which is significantly different. Though the percent of flies displaying incomplete pump is similar between nompC mutant and wild-type files, you can find it quite different in video 1 and 5.

Line 94: Should read: “while for foods with certain viscosity, the pump of Tmc or piezo mutants might"What evidence is there for weakened muscle motion? The phenotypes of all three mutants is quite similar, so concluding that they have roles in initiation versus swallowing strength is not well supported -this would be better moved to the discussion since it is speculative.

Muscles are responsible for pumping the bolus from the mouth to the crop. In the case of Tmc or piezo mutants, as evidenced by incomplete filling for viscous foods (see Video 4), we speculate that the loss of sensory stimuli leads to inadequate muscle contraction. The phenotypes observed in Tmc and piezo mutants are similar yet distinct from those of the wild-type or nompC mutant, as shown in Video 1 and 4. The phrase "due to weakened muscle motion" has been removed for clarity.

Line 146: If md-L neurons are also labeled by this intersection, then you are not able to know whether the axons seen in the brain are from md-L or md-C neurons. Line 148: cutting the labellum is not sufficient to ablate md-L neurons. The projections will still enter the brain and can be activated with optogenetics, even after severing the processes that reside in the labellum.

Please refer to the responses for reviewer #1 (Public Review):” A major weakness of the paper…” and Figure 4.

Line 162: If the fly head alone is in saline, do you know that the sucrose enters the esophagus? The more relevant question here is whether the md-C neurons respond to mechanical force. If you could artificially inflate the cibarium with air and see the md-C neurons respond that would be a more convincing result. So far you only know that these are activated during ingestion, but have not shown that they are activated specifically by filling or emptying. In addition, you are not only imaging md-C (md-L is also labeled). This caveat should be mentioned.

We followed the methods outlined in the previous work (Chen et al., Cell Rep., 2019, PMID:31644916), which suggested that md-C neurons do not respond to sugars. While we aimed to mechanically stimulate md-C neurons, detecting signal changes during different steps of swallowing is challenging. This aspect could be further investigated in subsequent research with the application of adequate patch recording or two-photon microscopy (TPM).

Figure 3: It is not clear what the pie charts in Figure 3 A refer to. What are the three different rows, and what does blue versus red indicate?

Figure 3A illustrates three distinct states driven by CsChrimson light stimulation of md-C neurons, with the proportions of flies exhibiting each state. During light activation, flies may display difficulty in filling, incomplete filling, or a normal range of pumping. The blue and red bars represent the proportions of flies showing the corresponding state, as indicated by the black line.

Figure 4: Where are the example traces for J? The comparison in K should be average dF/F before ingestion compared with average dF/F during ingestion. Comparing the in vitro response to sucrose to the in vivo response during ingestion is not a useful comparison.

Please refer to the answers for reviewer (#2 question d).

**Reviewer #2 (Recommendations For The Authors):**
Suggested experiments that would address some of my concerns listed in the public review include:a) high resolution SEZ images of MN-LexA lines crossed to LexAop-GFP to demonstrate their specificityb) more detail on the P2X2 experiment. It is hard to make suggestions beyond that without first seeing the details.c) presenting average GCaMP traces for all calcium imaging resultsd) to rule out taste stimulation of md-C (Figure 4K) I would suggest performing more extensive calcium imaging experiments with different stimuli. For example, sugar, water, and increasing concentrations of a neutral osmolyte (e.g. PEG) to suppress the water response. I think that this is more feasible than trying to get an in vitro taste prep to be convincing.

Please refer to the responses for public review of reviewer #2.

**Reviewer #3 (Recommendations For The Authors):**
Below I list my suggestions as well as criticisms.(1) It would be excellent if the authors could demonstrate whether varying levels of food viscosity affect md-C activation.

That is a good point, and could be studied in future work.

(2) It is not clear whether an intersectional approach using TMC-GAL4 and nompC-QF abolishes labelling of the labellar multidendritic neurons. If this is the case, please show labellar multidendritic neurons in TMC-GAL4 only flies and flies using the intersectional approach. Along with this question, I am concerned that labellum-removed flies could be used for feeding assay.

Intersectional labelling using TMC-GAL4 and nompC-QF could not abolish labelling of the labellar multidendritic neurons (Author response image 4). Labellum-removed flies could be used for feeding assay (Figure 3—figure supplement 1B-C, video 5), but once LSO or cibarium of fly was damaged, swallowing behavior would be affected. Removing labellum should be very careful.

Author response image 4

(3) Please provide the detailed methods for GRASP and include proper control.

Please refer to the responses for public review of reviewer #1.

(4) The authors hypothesized that md-C sequentially activates MN11 and 12. Is the time gap between applying ATP on md-C and activation of MN11 or MN12 different?Please refer to the responses for public review of reviewer #3. The time gap between applying ATP on md-C and activation of MN11 or MN12 didn’t show significant differences, and we think the reason is that the ex vivo conditions could not completely mimic in vivo process.I found the manuscript includes many errors, which need to be corrected.(1) The reference formatting needs to be rechecked, for example, lines 37, 42, and 43.(2) Line 44-46: There is some misunderstanding. The role of pharyngeal mechanosensory neurons is not known compared with chemosensory neurons.(3) Line 49: Please specify which type of quality of food. Chemical or physical?(4) Line 80 and Figure 1B-D Authors need to put filling and emptying time data in the main figure rather than in the supplementary figure. Otherwise, please cite the relevant figures in the text(S1A-C).(5) Line 84-85; Is "the mutant animals" indicating only nompC? Please specify it.(6) Figure 1a: It is hard to determine the difference between the series of images. And also label filling and emptying under the time.(7) S1E-H: It is unclear what "Time proportion of incomplete pump" means. Please define it.(8) Please reorganize the figures to follow the order of the text, for example, figures 2 and 4(9) Figure 4A. There is mislabelling in Figure 4A. It is supposed to be phalloidin not nc82.(10) Figure 4K: It does not match the figure legend and main text.(11) Figure 4D and G: Please indicate ATP application time point.

Thanks for your correction and all the points mentioned were revised.

**Reviewer #4 (Recommendations For The Authors):**
The figures need improvement. 1A has tiny circles showing pharynx and any differences are unclear.The expression pattern of some of these drivers (Supplement) seems quite broad. The tmc nompC intersection image in Figure 1F is nice but the cibarium images are hard to interpret: does this one show muscle expression? What are "brain" motor neurons? Where are the labellar multi-dendritic neurons?Tmc nompC intersection image show no expression in muscles. Somata of motor neurons 12 or 11 situated at SEZ area of brain, while somata of md-C neurons are in the cibarium. Image of md-L neurons was posted in response for reviewer #3 (Recommendations For The Authors):Why do the assays alternate between swallowing food and swallowing water?

Thank for your suggestion, figure 1A has been zoomed-in. The Tmc nompC intersection image in Figure 2F displayed the position of md-C neurons in a ventral perspective, and muscles were not labelled. We stained muscles in cibarium by phalloidin and the image is illustrated in Figure 4A, while we didn’t find overlap between md-C neurons and muscles. Image of md-L neurons were posted as Author response image 4.

In the majority of our experiments, we employed water to test swallowing behavior, while we used methylcellulose water solution to test swallowing behavior of mechanoreceptor mutants, and sucrose solution for flies with md-C neurons expressing GCaMP since they hardly drank water when their head capsules were open.

How starved or water-deprived were the flies?

One day prior to the behavioral assays, flies were transferred to empty vials (without water or food) for 24 hours for water deprivation. Flies who could not survive 24h deprivation would be deprived for 12h.

How exactly was the pumping frequency (shown in Fig 1B) measured? There is no description in the methods at all. If the pump frequency is scored by changes in blue food intensity (arbitrary units?), this seems very subjective and maybe image angle dependent. What was camera frame rate? Can it capture this pumping speed adequately? Given the wealth of more quantitative methods for measuring food intake (eg. CAFE, flyPAD), it seems that better data could be obtained.How was the total volume of the cibarium measured? What do the pie charts in Figure 3A represent?

The pump frequency was computed as the number of pumps divided by the time scale, following the methodology outlined in Manzo et al., 2012. Swallowing curves were plotted using the inverse of the blue food intensity in the cibarium. In this representation, ascending lines signify filling, while descending lines indicate emptying (see Figure 2D, 3B). We maintain objectivity in our approach since, during the recording of swallowing behavior, the fly was fixed, and we exclusively used data for analysis when the Region of Interest (ROI) was in the cibarium. This ensures that the intensity values accurately reflect the filling and emptying processes. Furthermore, we conducted manual frame-by-frame checks of pump frequency, and the results align with those generated by the time series analyzer V3 of ImageJ.

For the assessment of total volume of ingestion, we referred the methods of CAFE, utilizing a measurable glass capillary. We then calculated the ingestion rate (nL/s) by dividing the total volume of ingestion by the feeding time.

The changes seem small, in spite of the claim of statistical significance.

The observed stability in pump frequency within a given genotype underscores the significance of even seemingly small changes, which is statistically significant. We speculate that the stability in swallowing frequency suggests the existence of a redundant mechanism to ensure the robustness of the process. Disruption of one channel might potentially be partially compensated for by others, highlighting the vital nature of the swallowing mechanism.

How is this change in pump frequency consistent with defects in one aspect of the cycle - either ingestion (activation) or expulsion (inhibition)?

Please refer to Figure 2, 3. Both filling and emptying process were affects, while inhibition mainly influences emptying time (Figure 1—figure supplement 1).

for the authors:Line 48: extensivelyLine 62 - undiscovered.Line 107, 463: multiLine 124: What is "dysphagia?" This is an unusual word and should be defined.Line 446: severeLine 466: in the cibarium or not?

Thanks for your correction and all the places mentioned were revised.